# Hybrid Modification of Unsaturated Polyester Resins to Obtain Hydro- and Icephobic Properties

**Rafał Kozera [1,2,\*], Bartłomiej Przybyszewski [1,2], Katarzyna Żołyńska [1], Anna Boczkowska [1,2], Bogna Sztorch [3] and Robert E. Przekop [3]**

[1] Faculty of Materials Science and Engineering, Warsaw University of Technology, ul. Woloska 141, 02-507 Warszawa, Poland; bartlomiej.przybyszewski.dokt@pw.edu.pl (B.P.); katarzyna.zolynska.dokt@pw.edu.pl (K.Ż.); anna.boczkowska@pw.edu.pl (A.B.)

[2] Technology Partners Foundation, ul. Pawinskiego 5A, 02-106 Warszawa, Poland

[3] Centre for Advanced Technologies, Adam Mickiewicz University in Poznań, ul. Uniwersytetu Poznańskiego 10, 61-614 Poznań, Poland; Bogna.Sztorch@amu.edu.pl (B.S.); robert.przekop@amu.edu.pl (R.E.P.)

\* Correspondence: Rafal.Kozera@pw.edu.pl

**Abstract:** Ice accumulation is a key and unsolved problem for many composite structures with polymer matrices, e.g., wind turbines and airplanes. One of the solutions to avoid icing is to use anti-icing coatings. In recent years, the influence of hydrophobicity of a surface on its icephobic properties has been studied. This solution is based on the idea that a material with poor wettability maximally reduces the contact time between a cooled drop of water and the surface, consequently prevents the formation of ice, and decreases its adhesion to the surface. In this work, a hybrid modification of a gelcoat based on unsaturated polyester resin with nanosilica and chemical modifiers from the group of triple functionalized polyhedral oligomeric silsesquioxanes (POSS) and double organofunctionalized polysiloxanes (generally called multi-functionalized organosilicon compounds (MFSC)) was applied. The work describes how the change of modifier concentration and its structural structure finally influences the ice phobic properties. The modifiers used in their structure groups lowered the free surface energy and crosslinking groups with the applied resin, lowering the phenomena of migration and removing the modifier from the surface layer of gelcoat. The main studies from the icephobicity point of view were the measurements of ice adhesion forces between modified materials and ice. The tests were based on the measurements of the shear strength between the ice layer and the modified surface and were conducted using a tensile machine. Hydrophobic properties of the obtained nanocomposites were determined by measurement of the contact angle and contact angle hysteresis. As the results of the work, it was found that the modification of gelcoat with nanosilica and multi-functionalized silicone compounds results in the improvement of icephobic properties when compared to unmodified gelcoat while no direct influence of wettability properties was found. Ice adhesion decreased by more than 30%.

**Keywords:** icephobicity; hydrophobicity; ice adhesion; unsaturated polyester resin; nanosilica; multi-functionalized silicone compounds (MFSC)

## 1. Introduction

In recent years, increasingly, attention has been paid to the fight against ice accumulation and ice adhesion to various construction surfaces, including polymer composite constructions. These issues are still a key and unresolved problem for aircraft leading edges, turbojet engine blades, radars, water ships, wind turbines, building structures, or electrical installations. Ice build-up on wind turbines not only affects their energy efficiency but also affects their control and monitoring, causes electrical

and mechanical failures, as well as is a threat to human safety. The location of wind turbines in 20% is placed in areas with difficult weather conditions. As a result, the power losses resulting from icing can be up to 50% per year [1]. Currently, two main directions of icing reduction are known. The first one is active systems that include chemical, electrical, thermal, and mechanical ice removal methods, which are used in industries around the world. The second direction is the use of surfaces/coatings exhibiting anti-icing properties (passive systems). However, these surfaces are known for the time being mainly from literature reports. In the case that they would be durable, such passive systems could be a very good alternative to uneconomic and environmentally polluting active de-icing methods.

The design of high-performance, effective, and durable ice-phobic coatings remains a challenge. Coatings that exhibit anti-icing properties should show a low freezing temperature, low ice adhesion, low ice accretion rate, and long-term durability. The ice adhesion of polymer coatings varies between 150 kPa and 500 kPa [2]. However, these values should be much lower so that the coating can be called icephobic. In 2016 [3], it was possible to reduce ice adhesion below 10 kPa for the first time by enabling interfacial slippage on the ice surface without losing the durability of the coating. So far, the development of icephobic surfaces has been based on four basic types of surfaces: superhydrophobic/hydrophobic, slippery liquid infused porous surfaces (SLIPS) [4,5], hydrated and nonfrozen [6], and stress-localized [7]. However, all these solutions do not provide complete and permanent de-icing protection [8]. Recently, the most popular coatings tested for anti-icing properties are those showing hydrophobic and superhydrophobic properties. Superhydrophobic/hydrophobic surfaces can be fabricated by nano-/microstructuring the surface texture or by changing the chemical composition and giving the surface low surface energy (by using low surface energy materials). These hierarchical textures are capable of effectively capturing air and of creating air bags. This effect weakens the interaction of the cooled water drop with the structure's surface as well as reduces the surface and contact time between them and, consequently, reduces ice build-up [9]. Mechanisms of action of nano-/microstructures on wetting properties were discovered by studying phenomena occurring in nature, such as droplet behaviour on a lotus leaf or gecko's finger [10].

One way to give the polymer surfaces a micro/nanostructure and at the same time hydrophobic properties is to modify them with nanoparticles. An example of such nanoparticles is nano-$SiO_2$. Previously, the hydrophobic/hydrophobic properties of nanocomposite coatings containing nanosilica were demonstrated several times [11–13]. In 2015, Conradi et al. [14] studied the influence of nanosilica on the properties of epoxy resin. They showed that nanosilica increases the contact angle of the surfaces. Besides improving hydrophobic properties, nanosilica also has a positive effect on the corrosion resistance [14] and mechanical properties of resins [15]. Another approach is to chemically modify the composite in the bulk and on the surface with compounds that improve hydro-/icephobic properties by reducing surface energy, thereby preventing the adhesion of ice or water to the surface [16]. From a chemical point of view, modifiers that reduce the adhesion of ice and water to the surface of materials can be divided into the following groups: hybrid sol-gel coatings [17], hydrocarbon waxes [18], organofunctional silanes [19], fluorosilanes [20], alkylsilanes [21], polysiloxanes including dimeticones [22] and functionalized polysiloxanes [23], silsesquioxanes [24], silicone emulsions [25], perfluorinated resins [26], derivatives of high molecular weight carboxylic acids [27], and long chained amines [28]. Both icephobic and super-hydrophobic properties depend largely on the roughness of the surface and its surface energy. Chemical modifiers can affect the ability not only to repel water molecules from the surface but also to prevent condensation inside the structure and to significantly reduce the free energy of the surface, which contributes to reducing the effect of ice adhesion [29]. One of the most common modifiers are organosilicon compounds, mainly polysiloxanes [30], silanes including fluorinated silanes [31], or organofunctional silsesquioxanes [32,33]. Each of the mentioned groups are characterized by a specific structure and the ability to attach to the molecule functional groups, both organic, giving them the desired properties, and silanol ones, enabling connection with the polymer matrix. Polysiloxanes themselves are often used as surface modifiers towards icephobic or hydrophobic properties, but their additional functionalisation

contributes to both an increase in anti-adhesive properties and a reduction in free energy as well as an increase in viscoelasticity [34].

Polyhedral oligomeric silsesquioxanes and spherosilicates with a cubic well-defined structure are a unique group of organosilicon compounds that are modifiers of plastic materials that provide new properties or improve functional properties. Silsexioxanes play the role of modifiers for a wide group of materials, with the possibility of attaching various substituents to their inorganic siloxane cores, both inert, allowing for the improvement of ice and hydrophobic properties, as well as thermal, mechanical, and reactive groups enabling the attachment of silsesquioxanes with a polymer matrix [35]. Moreover, due to the presence of a strongly reactive Si–H bond, polysiloxanes, spherosilicates, and silsesquioxanes are functionalized in a hydrosilylation reaction with olefins containing various functional groups, which enables wide control of the properties of the modified composites [36]. One of their greatest advantages is the ability to attach as many as 8 different functional groups to the core, which control the properties of the modified material. Therefore, scientists have focused on the use of silsesquioxanes with up to 2 types of functional groups as hydrophobicity modifiers [17]. The use of 3 different functional groups is a novel approach.

There are many reports of improvement of icephobic properties by using surfaces with a contact angle greater than 90° (hydrophobic surface) or 150° (superhydrophobic surfaces) [37–39]. Mishchenko et al. have proven that a supercooled drop on superhydrophobic surfaces can endure negative conditions up to −30 °C [40]. However, there are also studies that give contradictory results. Ling et al. have proven that ice adhesion on a two-scale superhydrophobic surface can be up to 67% higher than on a polished copper surface [41]. Two contradictory relationships result from the fact that wettability at sub-zero temperatures differs from wettability at room temperature. The lack of improvement of icephobicity is related to the condensation of moisture and the deposition of frost in the micro/nanostructure at low temperature. These phenomena increase the contact area between the surface of the structure and the cooled water drop, which cause the ice to anchor and increase the heat transfer rate. As a result, the ice nucleation rate and the adhesion of the ice increase [9].

In this work, an attempt was made to develop a material with a surface exhibiting anti-icing properties by modifying a gelcoat based on unsaturated polyester resin with nanosilica powder and/or in-house synthesized chemical modifiers from the group of polyhedral oligomeric silsesquioxanes and double functionalized polysiloxanes (multi-functionalized organosilicon compounds (MFSC)). The investigated type of resin is widely used in wind turbine production. It is also one of the most commonly used resins for polymer composites [42–48]. POSSs, on the other hand, were supposed to give hydrophobicity in bulk, thanks to their highly hydrophobic peripheral organic groups. The anti-icing properties were determined by the ice adhesion strength test. The aim of this work was also to observe the relationship between hydrophobicity and icephobicity of the surface, as this relationship has not been fully understood so far.

## 2. Materials and Methods

### 2.1. Materials

The starting material used in the present studies was a gelcoat based on unsaturated polyester resin (UPR), Arctic-Gelcoat-ISO-S from BÜFA (Germany). It is a pre-accelerated gelcoat with a spraying consistency. The resin base is a pure isophthalic acid resin dissolved in styrene and HEMA (2-hydroxyethyl methacrylate). BÜFA®-Arctic-Gelcoat-ISO-S is suitable for moulded parts that are subjected to heavy weathering or normal hydrolysis loads. The nanosilica S5130 of 7 nm particle size used for modification was purchased from Sigma-Aldrich (United States).

The chemicals for modifier synthesis were purchased from the following sources: silicon compounds, polymethylhydrosiloxanes, and trimethylsiloxy terminated (PMHS) from Linegal Chemicals; octahydrospherosilicate, olefins (vinyltrimethoxysilane), hexene, octene, and 1H,1H,2H-perfluoro-1-decene from Linegal Chemicals; solvent toluene from Avantor Performance Materials

Poland S.A.; and chloroform-d, toluene-d8, and Karstedt catalyst from Sigma Aldrich. Toluene was dried and purified with the MB SPS 800 Solvent Drying System and stored under argon atmosphere in Rotaflo Schlenk flasks. Octahydrospherosilicate was prepared according to a literature procedure [48].

### 2.2. Synthesis of Organosilicon Precursors and Chemical Modifiers

#### 2.2.1. Synthesis of Poly((methyloctylsiloxane)-co-(methyl(trimethoxyethyl)siloxane)), Trimethylsilyl Terminated (MFSC 1)

Vinyltrimethoxysilne (0.168 mol) and octene (0.336 mol) in molar ratio 1:2 were added to the solution of PMHS (30 g, 0.504 mol) in toluene. The mixture was constantly stirred and heated to 70 °C. Then, it was added to the system in an amount which varied from $8 \times 10^{-5}$ eq. of the Karstedt catalyst. Reaction mixture was heated in reflux and stirred until the full conversion of Si–H was detected by $^1$H NMR and FT-IR. FT-IR analysis was performed every 6 h. Complete conversion of the substrates was observed after 48 h.

#### 2.2.2. Synthesis of Bis(3,3,4,4,5,5,6,6,7,7,8,8,9,9,10,10-Hexadecafluorodecyldimethylsiloxy)bis(hexyl-dimethylsiloxy) Tetrakis((trimethoxysilyl)ethyldimethylsiloxy)pentacyclo [9.5.1.13,9.15,15.17,13] octasiloxane (MFSC 2)

Vinyltrimethoxysilne (0.098 mol), 1H,1H,2H-Perfluoro-1-decene (0.0491 mol), and hexene (0.0491 mol) were added to the solution of octaspherosilicate (25 g, 0.196 mol) in toluene. The mixture was constantly stirred and heated to 60–70 °C. Then, it was added to the system in an amount which varied from $8 \times 10^{-4}$ of Pt. The reaction mixture was heated in reflux and stirred until the full conversion of Si–H was detected by $^1$H NMR and FT-IR. FT-IR analysis was performed every 6 h. Complete conversion of the substrates was observed after 24 h.

### 2.3. Preparation of Samples

Ten samples of a UPR-based gelcoat containing POSS and/or nanosilica were fabricated. One type of nanosilica ($nSiO_2$) powder (of 7 nm particle size) and two types of in-house synthesized POSS (marked as MFSC and MFSC 2) were used. Nanosilica was added to the gelcoat in the amount of 1 wt.%, and chemical modifiers were added in the amounts of 2 wt.% and 5 wt.%. The composition of the prepared samples and the amount of modifiers used are shown in Table 1. The sample preparation procedure was as follows. Nanosilica and/or chemical modifiers were added to the gelcoat in appropriate quantities and then stirred manually. Samples that contained nanosilica before modification were additionally treated with ultrasound to break the agglomerates and to even disperse the particles (process parameters were temperature 45 °C, time of processing 20 min, impulse duration 9 s of work and 2 s of break, and amplitude 39%). Furthermore, hardener Butanox M-50 was added in 1 wt.% quantity. Modified gelcoats were cast on two glass plates previously wrapped with anti-adhesive foil to avoid the additional influence of anti-adhesive liquids on the surface. Plastic frames of 200 × 100 mm were used to determine the size and thickness of the samples. The sample thickness was adjusted to 1 mm. In each case, two plates were fabricated to obtain samples with the same modified surface on both sides. On the half of the area limited by the frames, carbon fibre textiles were placed to reinforce the mounting area of the upper grip in the testing machine. The plates were cured for 30 min. After initial curing, the laminates were joined and glued by overlapping them to obtain a smooth surface of the modified gelcoat on both sides. The laminates were loaded and cured at room temperature for 24 h. Finally, the fabricated gelcoat plates were cut into samples with the size of 25 × 100 mm for ice adhesion testing. For the rest of the tests, samples with dimensions of 50 × 50 mm were cut.

**Table 1.** Composition of the prepared unsaturated polyester resin (UPR)-based gelcoat samples.

| Sample No. | Nanosilica | MFSC Type |
|:---:|:---:|:---:|
| 1 | - | - |
| 2 | $nSiO_2$ | - |
| 3 | - | MFSC 1/2 wt.% |
| 4 | - | MFSC 1/5 wt.% |
| 5 | $nSiO_2$ | MFSC 1/2 wt.% |
| 6 | $nSiO_2$ | MFSC 1/5 wt.% |
| 7 | - | MFSC 2/2 wt.% |
| 8 | - | MFSC 2/5 wt.% |
| 9 | $nSiO_2$ | MFSC 2/2 wt.% |
| 10 | $nSiO_2$ | MFSC 2/5 wt.% |

MFSC: multi-functionalized organosilicon compounds.

### 2.4. NMR and FTIR Analysis

$^{1}$H and $^{13}$C{$^{1}$H} NMR spectra were recorded on a Bruker Ultrashield 300 MHz operating at 300 and 75 MHz, respectively. $^{29}$Si{$^{1}$H} NMR spectra were recorded on a Bruker ascended 400 MHz Nanobay operating at 79 MHz, and Fourier transform infrared (FT-IR) spectra were recorded on a Nicolet iS50 Fourier transform spectrophotometer (Thermo Fisher Scientific) equipped with a diamond ATR unit with a resolution of 0.09 cm$^{-1}$. The spectra were collected in the 400–4000 cm$^{-1}$ range, and 16 scans were collected for each spectrum.

### 2.5. Determination of Roughness

The roughness tests were performed using a contact roughness meter (Surftest SJ-310 from Mitutoyo). Parameter such as the arithmetic mean of the roughness profile deviation ($R_a$) was measured. The measuring length was 10 mm, and the speed of move was 0.5 mm/s. The parameter results are the averaged values of three different measuring points on each surface.

### 2.6. Hydrophobicity Measurements

The wettability of the surfaces was tested by measuring the contact angle (CA) and by advancing (ACA) and receding (RCA) contact angles using a goniometer OCA15 (DataPhysics Instruments, Germany) with OCA software. The contact angle hysteresis (CAH) was determined by calculating the difference between the advancing and receding contact angles. A 5 µL droplet was used. The CA results are the averaged values of five different measuring points on each surface.

### 2.7. Ice Adhesion Measurements

The tests of ice adhesion to the surface of nanocomposite samples were based on measurements of the shear strength between the ice layer and the modified surface using the universal testing machine Zwick/Roel Z050. The geometry of the used samples is shown in Figure 1a. The samples were placed in specially designed metal holders. After water was poured into the holder, the samples were frozen at −10 °C for 24 h. After freezing, the samples on the holder were mounted in the testing machine in the bottom holder (Figure 1b). The area of the sample, which was previously reinforced with carbon fibre textiles, was mounted in the upper grip of the machine. During measurement, the maximum force was measured until the sample was released from the metal holder. The pull-out speed was 1 mm/min. The tests of ice adhesion were carried out at room temperature at a relative humidity of 50%.

In this work, ice adhesion is presented as the ratio of the maximum force obtained to the area of the contact surface. Six samples of each material were tested.

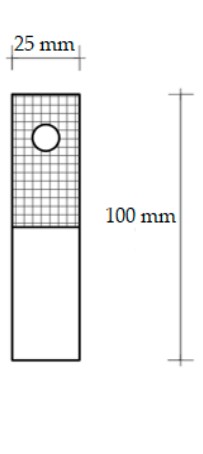
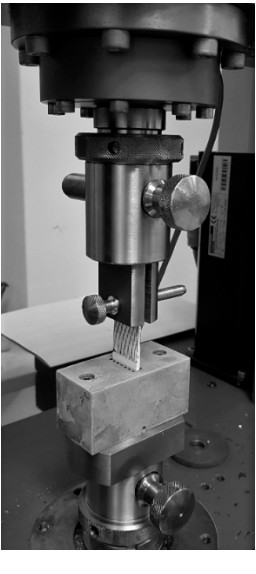

(**a**)                                                                                    (**b**)

**Figure 1.** (**a**) Sketch of the sample for ice adhesion testing and (**b**) testing machine with a mounted sample.

## 3. Results

### 3.1. Characterization of Organosilicon Modifiers

The organosilicon compounds were prepared according to the synthesis procedure in Section 2.2. They were investigated by NMR and IR spectroscopy to prove the completion of reactions (the disappearance of the characteristic signal was observed at 2141 and 889 cm$^{-1}$ due to stretching and bending of the Si–H group, respectively). The hydrosilylation reaction was complete for both products in c.a. 99%. The structure and purity of the modifiers obtained were confirmed by NMR analysis. It was observed that the hydrosilylation of olefins proceeded selectively to the $\beta$-isomer, except for trimethoxysililethyl, where the formation of $\alpha$-isomers was also observed in a proportion of 34% for polymethylhydrosiloxanes trimethylsiloxy terminated (Figure 2) and 20% for octaspherosilicate (Figure 3).

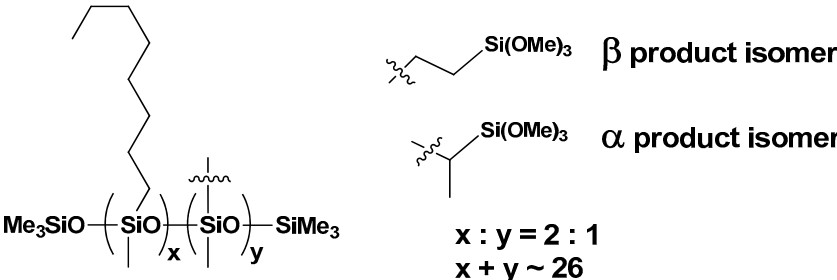

**Figure 2.** Structure of poly((methyloctylsiloxane)-co-(methyl(trimethoxyethyl)siloxane)), trimethylsilyl terminated.

The purity and chemical structure of the compound were confirmed by NMR spectroscopy, and the following signals were assigned:

$^1$H NMR    (400 MHz, CDCl$_3$): δ (ppm) = 3.56 (s, OMe), 1.30−1.27 (m, octyl -CH$_2$-), 1.09 (d, J = 7.5Hz, trimethylsiloxy alpha product), 0.88 (t, J = 6.2Hz, octyl –CH$_3$), 0.61–0.50 (m, SiCH$_2$ CH$_2$Si, SiCH(CH$_3$)Si), 0.14, 0.08, 0.07, 0.04 (s, SiMe$_2$, SiMe$_3$);

$^{13}$C NMR    (101 MHz, CDCl$_3$): δ (ppm) = 50.65 (OMe), 33.31, 31.83, 23.18, 23.06, 22.80, 17.84, 14.25 (octyl chain), 8.42 (Si–CH$_2$CH$_2$–Si), 1.98, 1.58, 0.64, −0.22, −1.14 (Si–CH$_2$CH$_2$–Si, SiMe$_2$, SiMe$_3$); and

$^{29}$Si NMR　(79,5 MHz, CDCl$_3$): δ (ppm) = −21.22, −22.37, (−22.82) (SiMe, SiMe$_3$), −38.00, −41.54 (OSi(OMe)$_3$).

**Figure 3.** Structure of bis(3,3,4,4,5,5,6,6,7,7,8,8,9,9,10,10-hexadecafluorodecyldimethylsiloxy) bis(hexyl dimethylsiloxy)tetrakis((trimethoxysilyl)ethyldimethylsiloxy)pentacyclo[9.5.1.13,9.15,15.17,13]octasiloxane.

The selectivity of the reaction, determined by NMR spectroscopy, shows that the product contains a mixture of the trimethoxysililethyl α and β group in a 20:80 ratio.

The purity and chemical structure of the compound were confirmed by NMR spectroscopy, and the following signals were assigned:

$^1$H NMR　(400 MHz, CDCl$_3$): δ (ppm) = 3.56 (s, 36H, OMe), 2.17–1.99 (m, 4H, –CF$_2$CH$_2$CH$_2$–), 1.32–1.25 (m, 16H, hexyl –CH$_2$–), 1.12 (d, J = 7.5Hz, trimethylsiloxy alpha product), 0.90–0.84 (m, 10H, hexyl –CH$_3$, –CF$_2$CH$_2$CH$_2$–), 0.60–0.56 (m, SiCH$_2$CH$_2$Si, SiCH$_2$–, trimethylsiloxy alpha product SiCH(CH$_3$)Si), 0.25 (s, 12H, fluorodecyl SiMe$_2$), 0.14, 0.12 (s, 36H, SiMe$_2$).

$^{13}$C NMR　(101 MHz, CDCl$_3$): δ (ppm) = 50.73, 50.66, 50.62 (OMe), 31.13, 31.71, 25.13, 23.04, 23.01, 22.98, 17.74, 14.23, 14.18, (hexyl chain, –CF$_2$CH$_2$–), 8.60 (Si–CH$_2$CH$_2$–Si), 7.34, 7.20, 5.27, 5.23 (SiCH(CH$_3$)Si), 0.42 (−1.60) (Si–CH$_2$CH$_2$–Si, SiMe$_2$, SiMe$_3$);

$^{29}$Si NMR　(79,5 MHz, CDCl$_3$): δ (ppm) = 13.75–13.06 (SiMe$_2$), −41.68, −42.84 (OSi(OMe)$_3$), −109.05 (cage).

### 3.2. Roughness of Surface

Tables 2 and 3 show the results of roughness. Table 2 shows all results for samples with the addition of modifier marked as MFSC 1, while Table 3 shows the results for samples with modifier marked as MFSC 2. Both tables also show the results of the reference sample and the sample modified only with nSiO$_2$ for comparison purposes.

**Table 2.** Description and roughness of the samples with multi-functionalized organosilicon compound (MFSC) 1.

| Sample | Nanosilica Used | MFSC Used | WCA (º) | CAH (º) | Ra (μm) |
|--------|-----------------|-----------|---------|---------|---------|
| 1 | - | - | 90 ± 1 | 34 ± 1 | 0.31 ± 0.09 |
| 2 | nSiO$_2$ | - | 86 ± 3 | 39 ± 3 | 0.41 ± 0.10 |
| 3 | - | MFSC 1/2 wt.% | 92 ± 2 | 25 ± 2 | 0.59 ± 0.06 |
| 4 | - | MFSC 1/5 wt.% | 104 ± 3 | 22 ± 1 | 0.39 ± 0.08 |
| 5 | nSiO$_2$ | MFSC 1/2 wt.% | 96 ± 3 | 32 ± 1 | 0.36 ± 0.08 |
| 6 | nSiO$_2$ | MFSC 1/5 wt.% | 101 ± 3 | 32 ± 1 | 1.14 ± 0.14 |

**Table 3.** Description and roughness of the samples with MFSC 2.

| Sample | Nanosilica Used | MFSC Used | WCA (°) | CAH (°) | Ra (μm) |
|--------|-----------------|-----------|---------|---------|---------|
| 1 | - | - | 90 ± 1 | 34 ± 1 | 0.31 ± 0.09 |
| 2 | nSiO$_2$ | - | 86 ± 3 | 39 ± 3 | 0.4 ± 0.10 |
| 7 | - | MFSC 2/2 wt.% | 106 ± 3 | 29 ± 1 | 0.29 ± 0.04 |
| 8 | - | MFSC 2/5 wt.% | 100 ± 3 | 21 ± 1 | 0.92 ± 0.09 |
| 9 | nSiO$_2$ | MFSC 2/2 wt.% | 104 ± 2 | 30 ± 2 | 1.05 ± 0.18 |
| 10 | nSiO$_2$ | MFSC 2/5 wt.% | 102 ± 1 | 27 ± 1 | 0.87 ± 0.06 |

The value of $R_a$ for the reference sample is 0.31 μm. Roughness for the rest of the samples was varied between 0.29 μm and 1.14 μm. Based on the results collected in Tables 2 and 3, it is visible that samples containing the MFSC 2 chemical modifier in the amount of 2 wt.% without nSiO$_2$ recorded the lowest value. It is worth noting that, despite all samples being were cast on the glass covered with foil only, only the sample containing nSiO$_2$ and the sample with 2 wt.% of MFSC 2 maintained comparable roughness values to the reference samples, while in the remaining cases, the addition of MFSC 2 and nSiO$_2$ caused an increased surface roughness. Thus, it is suggested that the addition of MFSC 2 can lead to some technological problems during fabrication, e.g., hindered the escape of styrene from the bottom of the sample and in the effect of a rougher surface. On the other hand, the addition of MFSC 1 indicated no similar effect. Only the sample with 5 wt.% of MFSC 1 with nSiO$_2$ caused visibly higher roughness due to the same technological problems as in the case of MFSC 2, while the rest of the results stayed in the range of the reference sample.

*3.3. Wetting Properties*

Water contact angle (WCA) and contact angle hysteresis (CAH) values for the samples before and after modifications were collected in Tables 2 and 3. The reference sample exhibits a WCA of 90° and CAH equal to 34°. Modification shows a positive effect on the tested values. It is observed that water contact angle increased in all samples containing chemical modifiers, i.e., MFSC 1 and MFSC 2. The highest results of 102° and 104° were obtained for samples containing MFSC 2, while increase in this additive concentration from 2 wt.% to 5 wt.% showed no further growth of WCA but a slight decrease. On the contrary, the contact angle of samples after modification with MFSC 1 exhibited further increase from 96° to 101° with the increase in modifier concentration. It is worth noting that the sample containing only nanosilica (without chemical modifier) showed a water contact angle equal to 86°, i.e., lower in comparison to the reference sample (90°). Chemically modified samples exhibited CAH values between 21° and 30°, while a sample only with nanosilica additives exhibited a hysteresis value of 39°, i.e., higher than the reference sample. In general, all modified gelcoat samples recorded an increase in water contact angle values and a decrease in contact angle hysteresis compared to the reference the sample as well as sample containing only nSiO$_2$.

*3.4. Ice Adhesion Strength*

Table 4 presents the values of ice adhesion (IA) of samples containing chemical modifier MFSC 1 as well as chemical modifier MFSC 2 to determine the effect of type and amount of POSS on the anti-icing properties. A lower value of this parameter means a better icephobic performance of the tested surface. Tests of ice adhesion showed that all samples containing chemical modifiers showed improved icephobic properties in comparison with the reference sample and that the sample containing only nSiO$_2$ gained reduced ice adhesion values which varied between 233 kPa and 320 kPa.

**Table 4.** The ice adhesion of fabricated samples.

| Sample | Nanosilica Used | MFSC Used | IA (kPa) |
|--------|-----------------|-----------|----------|
| 1 | - | - | 346 ± 18 |
| 2 | $nSiO_2$ | - | 395 ± 13 |
| 3 | - | MFSC 1/2 wt.% | 250 ± 15 |
| 7 | - | MFSC 2/2 wt.% | 263 ± 10 |
| 4 | - | MFSC 1/5 wt.% | 271 ± 25 |
| 8 | - | MFSC 2/5 wt.% | 320 ± 12 |
| 5 | $nSiO_2$ | MFSC 1/2 wt.% | 233 ± 19 |
| 9 | $nSiO_2$ | MFSC 2/2 wt.% | 279 ± 33 |
| 6 | $nSiO_2$ | MFSC 1/5 wt.% | 318 ± 30 |
| 10 | $nSiO_2$ | MFSC 2/5 wt.% | 284 ± 12 |

The sample with only $nSiO_2$ reached 346 kPa; that means about 43% average higher ice adhesion than samples with chemical modifiers. The lowest values of ice adhesion (233 kPa) were noted for hybrid modification of gelcoat with $nSiO_2$ and 2 wt.% of the chemical modifier MFSC 1. Compared with the reference gelcoat, ice adhesion of this sample decreased by more than 30%. Moreover, it can be observed that the values of adhesion strength depend on the amount of modifier. The samples with both MFSC 1 and MFSC 2 with 2 wt.% concentration recorded lower values of ice adhesion than samples containing higher amounts of these modifiers (i.e., 5 wt.%) in comparison to the reference samples. In the case of samples with 5 wt.% concentration of additives, it is assumed that the positive synergic effect was suppressed by the influence of increased additive concentration, which led to ice adhesion improvement in the referred samples without $nSiO_2$. However, considering only the samples with chemical modifiers without nanosilica, it is noticeable that the MFSC 1 modifier gives better results in terms of icephobic properties (both at 2 and 5 wt.%).

## 4. Discussion

### 4.1. Roughness/Wettability

Figure 4 shows the relationship between the parameters determining surface hydrophobicity (WCA and CAH) in the function of roughness ($R_a$) for samples modified with MFSC 1 in comparison to a reference sample and to the sample with $nSiO_2$ only. Analysis of the graphs gives a clear indication that wettability of the produced gelcoats does not depend directly on roughness. For similar $R_a$ values, WCA and CAH values change by leaps and bounds from 86° to 104° and from 39° to 22°, respectively, changing the surface character of the samples from hydrophilic to hydrophobic. By the addition of $nSiO_2$, WCA decreases to 86°, just below the values of the reference sample. The sample with $nSiO_2$ and 5 wt.% of MFSC 1 showed the highest parameter Ra, which was very different from the others, while no significant increase in wettability parameters was found when compared to the sample with only MFSC 1, exhibiting lower $R_a$. The obtained relationships contradict the Cassie–Baxter wettability model, indicating the positive influence of chemical additives synthesized in this work on the wettability parameters except the sample containing only $nSiO_2$. In this point, it is worth reminding the reader that higher amounts of added chemical modifier (5 wt.%) in the presence of $nSiO_2$ caused technological problems in production and thus increased the surface development of samples.

Figure 5 presents the relationship between the parameters determining the surface hydrophobicity (WCA and CAH) in the function of roughness ($R_a$) for samples with MFSC 2. Similar to the previously described samples containing MFSC 1, in this case no clear correlation between these parameters can be observed. However, it is important to note that both samples containing 5 wt.% of MFSC 2 and $nSiO_2$ showed significantly higher Ra values in comparison to sample no. 1, 2, and 7, while the highest WCA values were recorded for the sample containing 2 wt.% of MFSC 2, with the roughness value staying in the range of the reference samples.

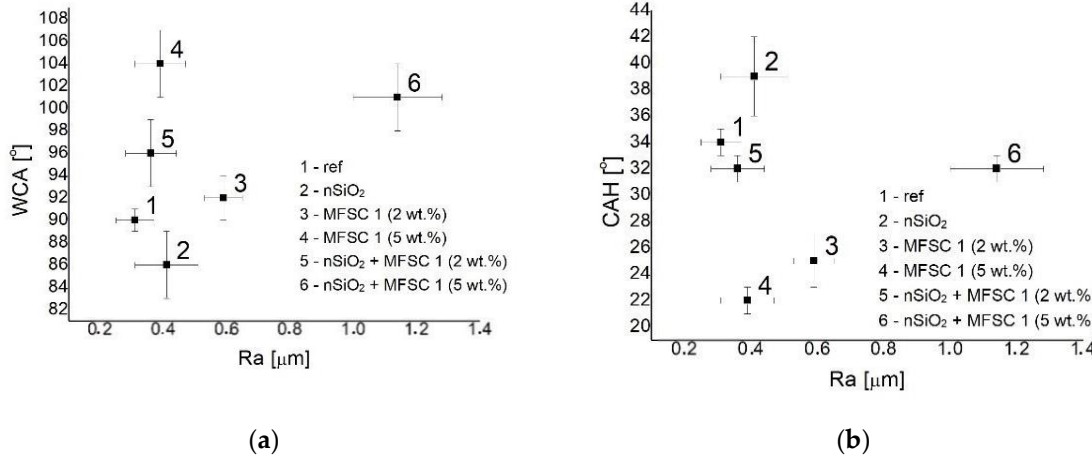

(**a**)                                                                                  (**b**)

**Figure 4.** Relationship of (**a**) the Water contact angle (WCA) and (**b**) contact angle hysteresis (CAH) in the function of $R_a$ for samples modified with MFSC 1.

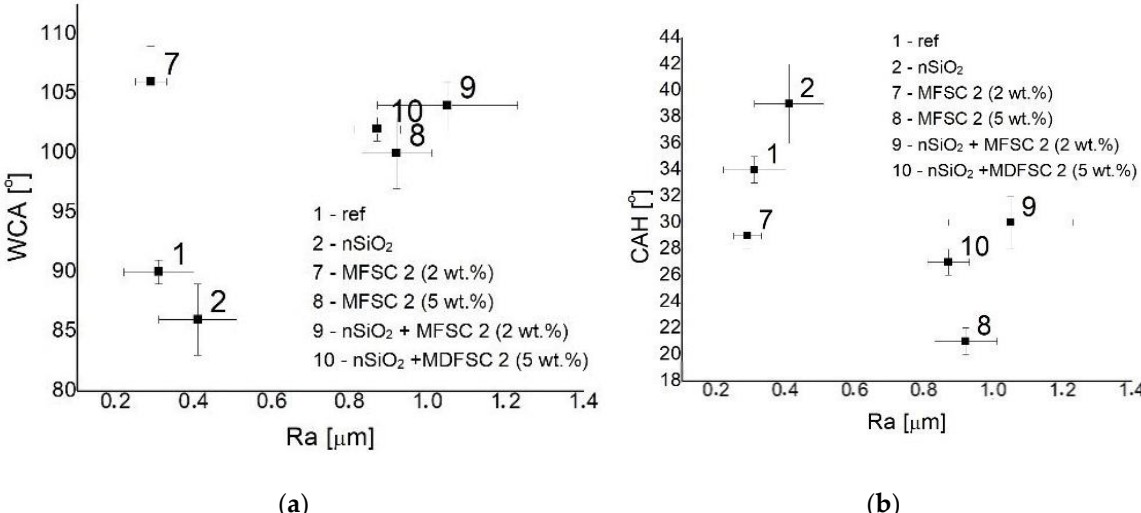

(**a**)                                                                                  (**b**)

**Figure 5.** Relationship of (**a**) the WCA and (**b**) CAH in the function of $R_a$ for samples modified with MFSC 2.

## 4.2. Roughness/Ice Adhesion

Based on the analysis of Figure 6, it can be concluded that the observed drop in ice adhesion values is clearly caused not by the roughness decrease but by the application of chemical modifiers and their positive influence on ice adhesion strength. For the sample containing MFSC 1, the lowest ice adhesion values (41% drop) were recorded for the sample containing 2 wt.% of additives while $R_a$ values stayed in the range of the reference sample. On the other hand, the significant increases of roughness over the reference samples for samples containing $nSiO_2$ with MFSC 1 (5 wt.%) do not lead to an increase in ice adhesion value. In the case of samples containing the MFSC 2 additive, the lowest ice adhesion values were recorded for the material with 2 wt.% of MFSC 2 with no $nSiO_2$ addition. Despite having similar roughness to the reference sample, the modified sample has a 24% lower ice adhesion than the unmodified gelcoat, which confirms the positive influence of chemical modification regardless of the roughness. Even a significant increase in roughness for sample no. 8–10 did not affect the ice adhesion values, which stayed below the reference sample and the sample containing only $nSiO_2$.

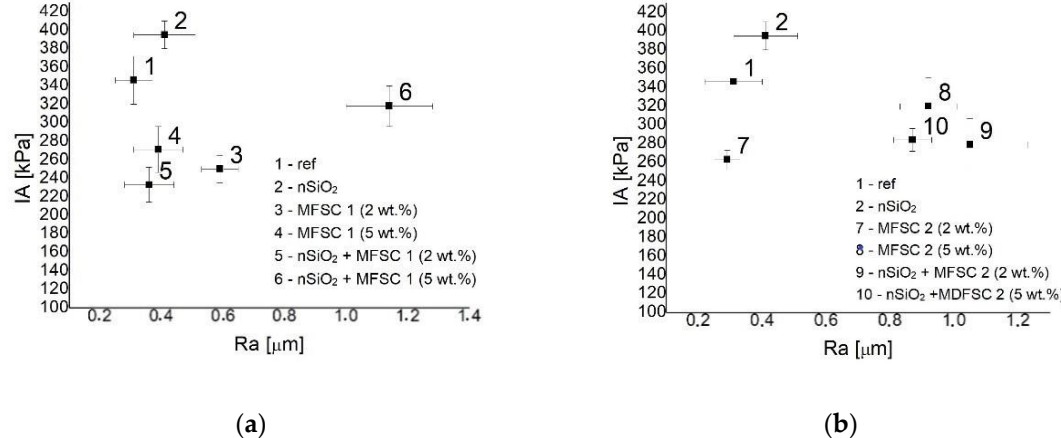

(**a**)             (**b**)

**Figure 6.** Relationship of IA (ice adhesion) as a function of $R_a$ for samples modified with (**a**) MFSC 1 and (**b**) MFSC 2.

### 4.3. Wettability/Ice Adhesion

The literature indicates many scientific studies which show a relationship between wettability and icephobicity of solids surface. The graphs collate the values of ice adhesion in the function of water contact angle (Figure 7a) and contact angle hysteresis (Figure 7b) for MFSC 1. Comparing those values with ice adhesion again, it is not possible to determine an unambiguous relation between these parameters. It can be noted that samples with the addition of 2 wt.% of MFSC 1 and the sample with $nSiO_2$ showed the lowest ice adhesion values among all tested samples despite their WCA not being the highest. Both samples had a slightly hydrophobic surface character, with WCAs of 92° and 96°, respectively. This means that WCA exhibits no direct influence on ice adhesion values and that higher contact angles do not provide a decrease in ice adhesion values. Hence, it indicates that the presence of the applied modifiers plays a main role in the ice adhesion mechanism. A similar situation was recorded for ice adhesion strength in the function of CAH. No direct correlation was found.

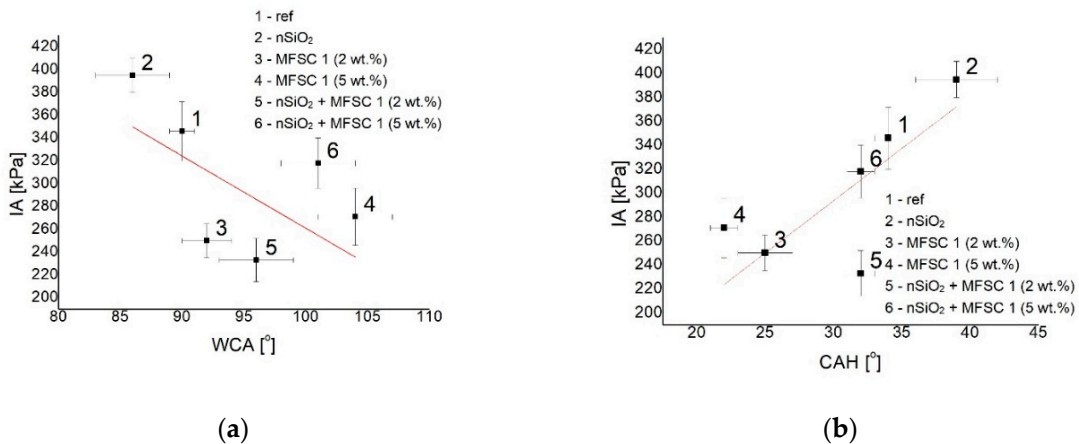

(**a**)             (**b**)

**Figure 7.** Relationship of IA as a function of (**a**) WCA and (**b**) CAH for samples modified with MFSC 1.

Figure 8 shows the relationship between wettability parameters and ice adhesion for samples with MFSC 2. In contrast to samples with modifier MFSC 1, a clear correlation between WCA and IA can be seen for samples with MFSC 2. As the WCA values increase, the ice adhesion of all samples decreases. The sample with the highest WCA of 106° has the lowest ice adhesion to the surface (reduced by 24% compared to the reference sample). In general, the lowest ice adhesion and the highest WCA were recorded for 2 wt.% of MFSC 2 with and without $nSiO_2$. On the other hand, no similar relationship

was recorded for CAH values. In this case, samples with the lowest CAH exhibited just a slight drop in ice adhesion in comparison to the reference samples.

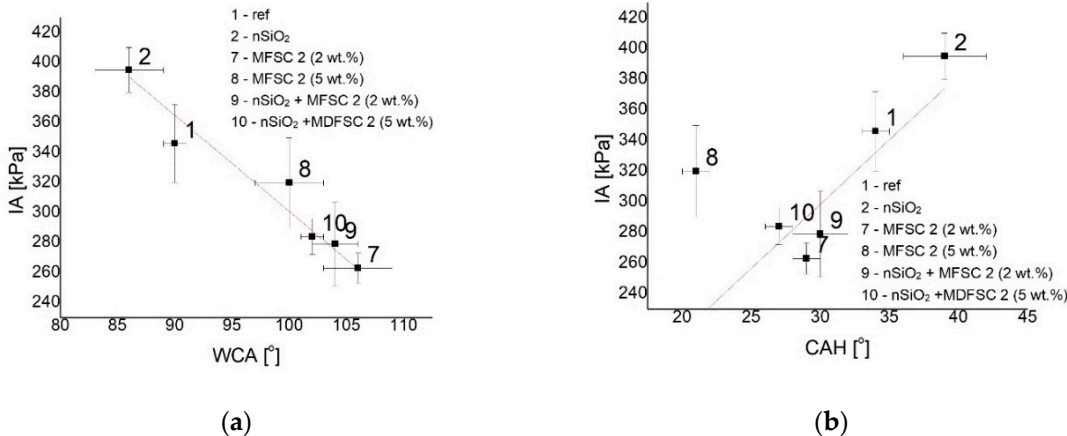

| (**a**) | (**b**) |

**Figure 8.** Relationship of IA as a function of (**a**) WCA and (**b**) CAH for samples modified with MFSC 2.

After analysing the results collected in Figure 9, it can be concluded that, except the samples modified only by $nSiO_2$, all tested samples showed an improvement in the icephobic properties compared to the reference non-modified sample. This improvement is associated with a decrease in the value of ice adhesion strength to the surface (by more than 30% compared to the reference sample).

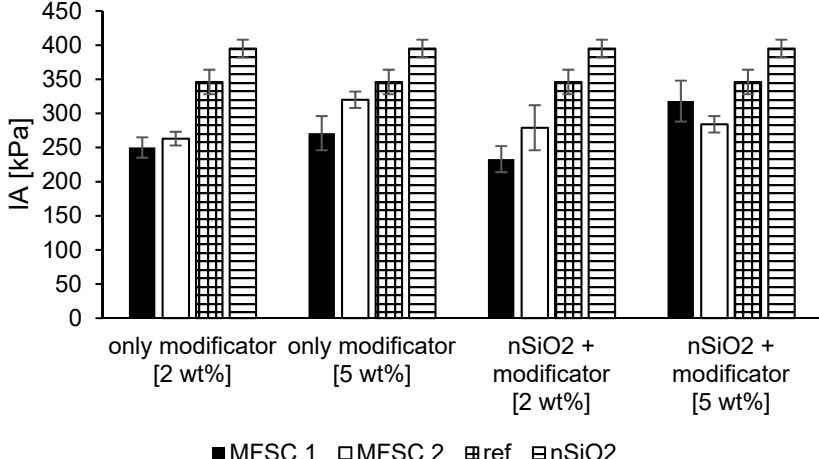

**Figure 9.** The ice adhesion of produced samples.

## 5. Conclusions

Summarizing, all modifications showed a positive effect on hydrophobic properties. It was observed that the water contact angle increased in all samples containing chemical modifiers, i.e., MFSC 1 and MFSC 2. The highest results were obtained for samples containing MFSC 2 except alkyl, alkoxy, and fluorinated functional groups; however, increase in this additive concentration from 2 wt.% to 5 wt.% showed no further increase in water contact angle. On the contrary, the contact angle of samples after modification with MFSC 1 (alkyl and alkoxy functional groups) exhibited further increase in water contact angle with an increase in modifier concentration up to 5 wt.%. On the other hand, the concentration of 5 wt.% of synthesized additives can lead to some technological problems during fabrication. The investigated relationships between roughness and wettability contradict the Cassie–Baxter wettability model, indicating the positive influence of chemical additives, especially with fluorinated groups synthesized in this work on wettability parameters.

Conducted tests on ice adhesion showed that all samples containing chemical modifiers indicated improved icephobic properties in comparison with the reference sample and the sample containing only nSiO$_2$. Moreover, during the investigation, it was found that ice adhesion does not depend on the value of water contact angles and contact angles hysteresis but on the types of chemical additives and their concentrations. The samples with MFSC 2 (alkyl and fluorinated functional groups) exhibited a direct correlation between WCA and IA (as the water contact angle increased, the ice adhesion decreased), while the samples with MFSC 1 exhibited no such correlation. However, it can be stated that the hydrophobicity of the surface is conducive to the improvement of anti-icing properties. The effect of a drop in ice adhesion values is clearly caused not by the roughness decrease but by the application of chemical modifiers with characteristic alkyl and fluorinated groups and their positive influence.

Comparing the two applied chemical modifiers, it can also be concluded that, in most cases, the addition of MFSC 1 was more beneficial by giving lower ice adhesion results than the MFSC 2 additive. It is also important to note that the sample with nanosilica showing the highest value of ice adhesion after chemical modification with spherosilicate compounds recorded a decrease in ice adhesion value and thus an improvement of the icephobic properties. The main element of novelty in the presented work is the use of multifunctional modifiers (both crosslinking and surface energy changing) in combination with modifier nano- and microstructures. Currently, there are no literature reports on the use of such compounds in icephobic structures.

**Author Contributions:** Conceptualization, R.K. and B.S.; methodology, B.P.; validation, R.E.P.; investigation, K.Ż.; supervision, A.B. All authors have read and agreed to the published version of the manuscript.

**Funding:** This research was funded by National Centre for Research and Development (NCBiR), grant number LIDER/16/0068/L-9/17/NCBR/2018.

**Conflicts of Interest:** The authors declare no conflict of interest.

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
