# Peer review of "Hybrid Modification of Unsaturated Polyester Resins to Obtain Hydro- and Icephobic Properties"

_processes, doi:10.3390/pr8121635_

Round 1
Reviewer 1 Report
The work entitled “Hybrid modification of unsaturated polyester resins to obtain hydro and icephobic properties” supposes an interesting work which deals with the development of hybrid coatings with icephobic properties through the formulation of commercial polyester resins with synthetic silsesquioxanes in combination with SiO2 nanoparticles.
The work is well structured and well explained, although some aspects could be improved. Following, the authors can find different suggestions and comments for the improvement of the submitted paper:
- As it can be inferred from the paper, the ice formation is undertaken previously to ice adhesion tests. Could it be posible to perform this stage in situ in the universal testing machine with a climatic chamber? This point coul help to obtain more comparable or reproducible results.
- It should be interesting to measure the durability of the icephobicity of the modified coatings after several icing-deicing cycles.
- A posible explanation of the reason why there is not correlation between ice adhesion with roughness, WCA or CAH should be included. Is there any chemical component behind this result? The measurement of the surface tension could give some enlightenment to this point.
- A comparison of the ice adhesion values obtained with other similar icephobic coatings should be included.
- The last sentence in conclusions “ synergic modification of the gelcoat combining nSiO2 and modifieres has brought the most favorable results” is not totally true. It´s clear from the different results that samples with only POSS gave positive results in terms of wettability, CHA or ice adhesion. However, when nSiO2 are mixed with these latter modified resins a detrimental result is observed, through the decrease of wettability, increase of CAH and at last the increase of ice adhesion, comparing with their counterpart formulations without silica nanoparticles although the global result is better than for the unmodified reference.
- In this regard, it would be interesting to study higher concentrations of nSiO2 in order to increase in a significant way the final roughness obtaining an effective increase of contact angle or a reduction of the final ice adhesion. Did you already study other concentrations of SiO2 nanoparticles?
Author Response
- As it can be inferred from the paper, the ice formation is undertaken previously to ice adhesion tests. Could it be possible to perform this stage in situ in the universal testing machine with a climatic chamber? This point could help to obtain more comparable or reproducible results.
A: Thank you for your valuable suggestion. We fully agree with you and we will take it into consideration during next tests which we are going to proceed. Nevertheless, we repeated few times our ice adhesion tests and we haven’t found any deviation resulting from methodology which we took so far.
- It should be interesting to measure the durability of the icephobicity of the modified coatings after several icing-deicing cycles.
A: Thank you for your valuable suggestion. We agree that this test will give large amount of interesting results and observations. Nevertheless, we are going to prepare separate paper only regarding those findings and thus in the present paper we decided to not include those results.
- A possible explanation of the reason why there is not correlation between ice adhesion with roughness, WCA or CAH should be included. Is there any chemical component behind this result? The measurement of the surface tension could give some enlightenment to this point.
A: Thank you for your valuable comment. We fully agree with you. The results fully confirm the current literature picture of issues related to icephobicity. The obtained results confirm that there are no unambiguous correlations between the contact angle for water and icephobicity. The phenomena related to the formation of the ice layer are much more complex due to the possible ways of ice crystallization on the surface of the material and the resulting interactions. At this stage of research, we were focused on a new generation of multi-functional chemical modifiers, and their functional groups (alkyl and fluorinated). Those chemical components i.e. functional groups are responsible for described relations (lines 442-475). During next publication which we are now preparing we intend to look deeper in those correlations and for sure we will perform surface tension tests at the direction indicated in your review. One more time thank you for this comment.
- A comparison of the ice adhesion values obtained with other similar icephobic coatings should be included.
A: Thank you for your valuable suggestion. For now there is no findings in the literature which could be directly comparable with our material unsaturated polyester resin, in more general way our results can be referred to ice adhesion values found in the lines 58-62.
- The last sentence in conclusions “ synergic modification of the gelcoat combining nSiO2 and modifieres has brought the most favorable results” is not totally true. It´s clear from the different results that samples with only POSS gave positive results in terms of wettability, CHA or ice adhesion. However, when nSiO2 are mixed with these latter modified resins a detrimental result is observed, through the decrease of wettability, increase of CAH and at last the increase of ice adhesion, comparing with their counterpart formulations without silica nanoparticles although the global result is better than for the unmodified reference.
A: Thank you for your valuable comment. We agree with you that it was misleading. We erased it from the text.
- In this regard, it would be interesting to study higher concentrations of nSiO2 in order to increase in a significant way the final roughness obtaining an effective increase of contact angle or a reduction of the final ice adhesion. Did you already study other concentrations of SiO2 nanoparticles?
A: Thank you for your comment. We have studied other concentrations of silica. The higher amount of this modifier increased the viscosity of the mixtures and as a result, it was not possible to produce samples correctly in most cases. In addition, gelcoat with more silica recorded higher ice adhesion. Additionally, in 2012 Sudirman et al. [Sudirman, M. Anggaravidya, E. Budianto, and I. Gunawan, “Synthesis and Characterization of Polyester-Based Nanocomposite,” Procedia Chem., vol. 4, pp. 107–113, 2012, doi: 10.1016/j.proche.2012.06.016.] produced a component system consisting of unsaturated polyester resin and nanosilica. They observed that the highest concentration of this nano-additive, at which its normal dispersion in the matrix can be achieved, is 1%. We also took this information into account when determining the amount of SiO2 used.
Reviewer 2 Report
In this work, the authors developed an anti-icing surface utilizing an unsaturated polyester resin modified with nano-scale silica powder and groups of polyhedral oligomeric silsesquioxanes (POSS). The authors then investigated the effect of water wettability and roughness on the ice adhesion. The results of this work appear to have a broader impact on anti-icing coatings.
Here I listed a few suggestions and comments.
- The literature review does not seem to be up to date. I suggest the authors to include more references as listed below.
- Current anti-icing papers by using similar approaches
“A predictive framework for the design and fabrication of icephobic polymers”, Science advances, 3(9), e1701617.
“Designing durable icephobic surfaces”. Science advances, 2(3), e1501496.
- The authors can include the following papers on the related applications of POSS:
“On‐demand separation of oil‐water mixtures”, Advanced materials, 24(27), 3666-3671.
“Hygro-responsive membranes for effective oil–water separation”, Nature communications, 3(1), 1-8.
- Nanoparticles can modify the polymer’s physicochemical properties:
“Self-healable superomniphobic surfaces for corrosion protection”, ACS applied materials & interfaces, 11(33), 30240-30246.
“Effect of nanoparticles on the anticorrosion and mechanical properties of epoxy coating” Surface and Coatings Technology, 204(3), 237-245.
- Sample preparation section: The authors limit the compositions of silica nanoparticles and chemical modifiers to 1 wt.%, 2 wt.%, and 5 wt.%. The authors can explain the rationale for this selection.
- Materials synthesis section: The authors need to clarify the time at which a full conversion of Si-H was observed.
- Ice adhesion measurement section: The authors need to elaborate on the conditions of the measurements (e.g., temperature, relative humidity, etc.)
- The WCA and CAH data are shown in Figures 4-6 and Table 2 and Table 3. The authors can remove one.
- Interestingly, adding silica particles resulted in a decrease in the average roughness of POSS1 sample (see Table 2). The authors can explain why?
- Line 344: The authors can elaborate on the ‘drop effect’.
- Wettability/ice adhesion section: The first paragraph concludes with no correlation between the ice adhesion and WCA/CAH while the next paragraph presents contradictory results. I may miss something. The authors can clarify the results.
- The size and geometry of the ice sample used for adhesion tests need to be clarified.
- Overall, the plots’ quality can be improved. Perhaps the authors can use a larger marker and/or title.

Author Response
- The literature review does not seem to be up to date. I suggest the authors to include more references as listed below.
Current anti-icing papers by using similar approaches
“A predictive framework for the design and fabrication of icephobic polymers”, Science advances, 3(9), e1701617.
“Designing durable icephobic surfaces”. Science advances, 2(3), e1501496.
The authors can include the following papers on the related applications of POSS:
“On‐demand separation of oil‐water mixtures”, Advanced materials, 24(27), 3666-3671.
“Hygro-responsive membranes for effective oil–water separation”, Nature communications, 3(1), 1-8.
Nanoparticles can modify the polymer’s physicochemical properties:
“Self-healable superomniphobic surfaces for corrosion protection”, ACS applied materials & interfaces, 11(33), 30240-30246.
“Effect of nanoparticles on the anticorrosion and mechanical properties of epoxy coating” Surface and Coatings Technology, 204(3), 237-245.
A: Thank you for your suggestions. We analysed literature which you recommended and decided to join to our literature review two articles as the most corresponding to subject of our work (paragraph lines 58-62, literature positions [2,3]).
- Sample preparation section:
The authors limit the compositions of silica nanoparticles and chemical modifiers to 1 wt.%, 2 wt.%, and 5 wt.%. The authors can explain the rationale for this selection.
A: Thank you for your comment. We have studied other concentrations of silica and DFSC compounds. The higher amount of SiO2 caused an increase in the viscosity of the mixtures and as a result it was not possible to produce samples correctly in most cases. In addition, a gelcoat with a higher silica content showed higher ice adhesion. In addition, in 2012, Sudirman et al. Sudirman, M. Anggaravidya, E. Budianto, and I. Gunawan, "Synthesis and Characterization of Polyester-Based Nanocomposites", Procedia Chem., vol. 4, p. 107-113, 2012, doi: 10.1016/j.proche.2012.06.016.] produced a component system consisting of unsaturated polyester resin and nanosilica. It was found that the highest concentration of this nanoadditive, at which its normal dispersion in the matrix can be obtained, is 1%. We also took this information into account when determining the amount of SiO2 used. The amount of DFSC compounds was determined, among others, by the cost of these modifiers. Moreover, the increase in the amount of DFSC did not result in any improvement of de-icing properties in comparison with lower amount equivalents. In further work, we plan to produce gelcoats with lower concentration of DFSC to reduce production costs as much as possible and to determine whether they have lower ice adhesion values.
- Materials synthesis section: The authors need to clarify the time at which a full conversion of Si-H was observed.
A: Thank you for your valuable suggestion, we added information in the paragraph (lines 165-166 and 175-176) with respect to your suggestion.
- Ice adhesion measurement section: The authors need to elaborate on the conditions of the measurements (e.g., temperature, relative humidity, etc.)
A: Thank you for your valuable suggestion, we added information in the paragraph (line 231-232) with respect to your suggestion.
- The WCA and CAH data are shown in Figures 4-6 and Table 2 and Table 3. The authors can remove one.
A: Thank you for your valuable suggestion, according your comment we decided to remove Figures 4 and 5, while keep Fig 6 because it facilitates discussion of results in discussion section.
- Interestingly, adding silica particles resulted in a decrease in the average roughness of POSS1 sample (see Table 2). The authors can explain why?
A: Thank you for your comment. In fact, average roughness of sample containing SiO2 and addition of 5% exhibit significantly higher roughness, probably due to addition of POSS1 additive can lead to some technological problems during fabrication, e.g., hindered the escape of styrene from the bottom of the sample and in effect of more rough surface. Complementary description in paper, lines 300-301.
- Line 344: The authors can elaborate on the ‘drop effect’.
A: Thank you for your comment. In fact phrase ‘drop effect’ was misleading. Correction is made in lines 382-383.
- Wettability/ice adhesion section: The first paragraph concludes with no correlation between the ice adhesion and WCA/CAH while the next paragraph presents contradictory results. I may miss something. The authors can clarify the results.
A: Thank you for your comment. The first paragraph refers to samples with modifier DFSC 1 (no correlation was observed between wettability and ice adhesion). The second paragraph refers to the samples with modifier DFSC 2 (in which correlations between wettability and ice adhesion have been observed).
- The size and geometry of the ice sample used for adhesion tests need to be clarified.
A: Thank you for your valuable suggestion. The geometry of the ice adhesion test samples was determined, among other things, by the dimensions of the testing machine on which they were performed. The geometry of the samples was created by the authors of the article taking into account the available equipment and the need to determine the ice adhesion. The key dimension in these tests was not the geometry of the samples, but the contact surface of the sample with ice, as it was this value that was used directly for the adhesion calculations.
- Overall, the plots’ quality can be improved. Perhaps the authors can use a larger marker and/or title.
A: Thank you for your valuable suggestion. We improved plots according to your suggestion.